# 3DITSCENE: EDITING ANY SCENE VIA LANGUAGE-GUIDED DISENTANGLED GAUSSIAN SPLATTING

**Qihang Zhang**
CUHK
qhzhang@link.cuhk.edu.hk

**Yinghao Xu**
Stanford
yhxu@stanford.edu

**Chaoyang Wang**
Snap Inc.
cwang9@snapchat.com

**Hsin-Ying Lee**
Snap Inc.
hlee5@snapchat.com

**Gordon Wetzstein**
Stanford
gordon.wetzstein@stanford.edu

**Bolei Zhou**
UCLA
bolei@cs.ucla.edu

**Ceyuan Yang**
ByteDance
limbo0066@gmail.com

Figure 1: **Image pairs edited by `3DitScene`.** Our method is capable of simultaneously handling various types of edits in both 2D and 3D spaces.

## ABSTRACT

Scene image editing is crucial for entertainment, photography, and advertising design. Existing methods solely focus on either 2D individual object or 3D global scene editing. This results in a lack of a unified approach to effectively control and manipulate scenes at the 3D level with different levels of granularity. In this work, we propose `3DitScene`, a novel and unified scene editing framework leveraging language-guided disentangled Gaussian Splatting that enables seamless editing from 2D to 3D, allowing precise control over scene composition and individual objects. We first incorporate 3D Gaussians that are refined through generative priors and optimization techniques. Language features from CLIP then introduce semantics into 3D geometry for object disentanglement. With the disentangled

Gaussians, `3DitScene` allows for manipulation at both the global and individual levels, revolutionizing creative expression and empowering control over scenes and objects. Experimental results demonstrate the effectiveness and versatility of `3DitScene` in scene image editing. Code is available at `https://github.com/zqh0253/3DitScene`.

# 1 INTRODUCTION

Editing scene images is of great importance in various fields, ranging from entertainment, professional photography and advertising design. Content editing allows to create immersive and captivating experiences for audiences, convey the artistic vision effectively and achieve the desired aesthetic outcomes. With the rapid development of deep generative modeling, many attempts have been made to edit an image effectively. However, they have encountered limitations that hindered their potential.

Previous methods primarily concentrate on scene editing in 2D image space. They commonly rely on generative priors, such as GANs and Diffusion Models (DM), and employ techniques like modification of cross-attention mechanisms (Hertz et al., 2022; 2023), and optimization of network parameters (Kim et al., 2022; Kawar et al., 2023; Ruiz et al., 2023; Gal et al., 2022; Chen et al., 2023b) to edit the appearance and object identity within scene images. While some efforts have been made to extend these methods to 3D editing, they ignore 3D cues and pose a challenge in maintaining 3D consistency, especially when changing the camera pose. Moreover, these approaches typically focus on global scenes and lack the ability to disentangle objects accurately, resulting in limited control over individual objects at the 3D level.

In order to edit any scene images and enable 3D control over both scene and its individual objects, we propose `3DitScene`, a novel scene editing framework which leverage a new scene representation, language-guided disentangled Gaussian Splatting. Concretely, the given image is first projected into 3D Gaussians which are further refined and enriched through 2D generative prior (Rombach et al., 2022; Poole et al., 2022). We thus obtain a comprehensive 3D scene representation that naturally enables novel view synthesis for a given image. In addition, language features from CLIP are distilled into the corresponding 3D Gaussians to introduce semantics into 3D geometry. These semantic 3D Gaussians help disentangle individual objects out of the entire scene representation, resulting in language-guided disentangled Gaussians for scene decomposition. They also allow for a more user-friendly interaction *i.e.,* users could query specific objects or interest via text. To this end, our `3DitScene` enables seamless editing from 2D to 3D and allow for modifications at both the global and individual levels, empowering creators to have precise control over scene composition, and object-level edits.

We dub our pipeline as `3DitScene`. Different from previous works that focus on addressing a single type of editing, `3DitScene` integrates editing requirements within a unified framework. Our teaser figure demonstrates the versatility of `3DitScene` by showcasing its application to diverse scene images. We have conducted evaluations of `3DitScene` under various settings, and the results demonstrate significant improvements over baseline methods.

# 2 RELATED WORK

**Image Editing with Generative Models.** The field of 2D image synthesis has advanced significantly with the development of generative models such as GANs (Karras et al., 2021; 2019) and diffusion models (Rombach et al., 2022; Song et al., 2020; Ho et al., 2020). Many studies capitalize on the rich prior knowledge embedded in generative models for image editing. Some endeavors utilize GANs for various image editing tasks, including image-to-image translation, latent manipulation (Shen et al., 2020; Yang et al., 2021; Zhu et al., 2020; Xu et al., 2021; Jahanian et al., 2019), and text-guided manipulation (Patashnik et al., 2021). However, due to limitations in training on large-scale data, GANs often struggle to perform well on real-world scene images. As diffusion models make notable progress, the community is increasingly focusing on harnessing the potent text-to-image diffusion model for real image editing (Kim et al., 2022; Kawar et al., 2023; Ruiz et al., 2023; Gal et al., 2022; Chen et al., 2023b; Hertz et al., 2022; 2023; Meng et al., 2021b;

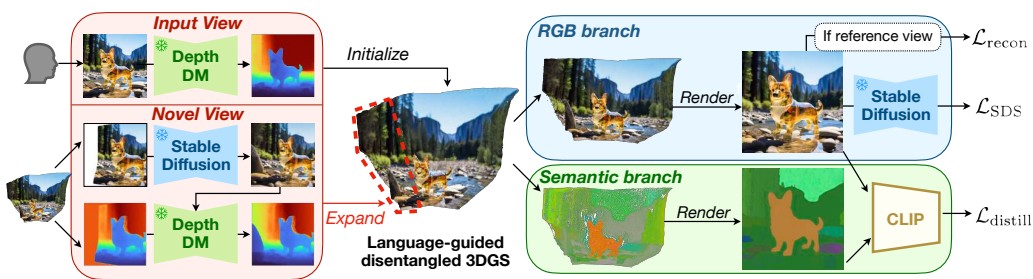

Figure 2: **3DitScene training pipeline.** Given input view, we first initialize 3DGS by lifting pixels to 3D space and then expand it over novel views by RGB and depth inpainting. Semantic features are then distilled into 3D Gaussians to achieve object-level disentanglement.

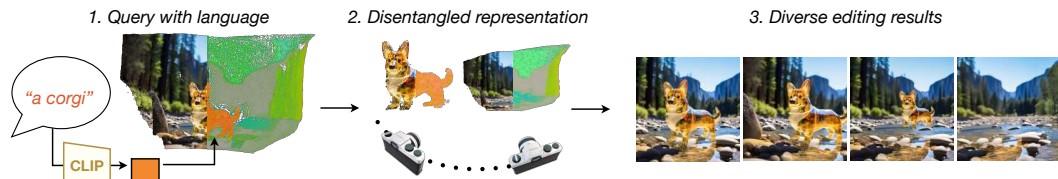

Figure 3: **3DitScene Inference pipeline.** User can query object of interest via language prompt. Enabled by the disentangled 3D representation, user can change camera viewpoint, and manipulate the object of interest in a flexible manner.

Su et al., 2022). However, these methods are confined to the 2D domain and are limited in editing objects within a 3D space. Concurrently, other research efforts (Yenphraphai et al., 2024a) attempt to address 3D-aware image editing, but they introduces inconsistency in the editing process, and cannot change the camera perspective of the entire scene. Wang et al. (2024a); Chen et al. (2024a); Ye et al. (2023); Palandra et al. (2024); Wu et al. (2024a); Wang et al. (2024b); Jaganathan et al. (2024) focus on editing with a given 3DGS scene, but is limited in the types of edits they support. In contrast, our model leverages an explicit 3D Gaussian to convert 2D images into 3D space while disentangling objects with language guidance. This approach enables our model not only to consistently perform 3D-aware object editing but also facilitates scene-level novel-view synthesis.

**Single-view 3D Scene Synthesis.** Among 3D scenes generation (Zhang et al., 2023b; Höllein et al., 2023; Chung et al., 2023; Chen et al., 2023a;c; Mao et al., 2023; Epstein et al., 2024), conditional generation on a single-view presents an unique challenge. Previous approaches address this challenge by training a versatile model capable of inferring a 3D representation of a scene based on a single input image (Wiles et al., 2020; Tucker & Snavely, 2020; Hu et al., 2021; Han et al., 2022; Flynn et al., 2019; Li et al., 2021; Hong et al., 2023; Yu et al., 2021). However, these methods demand extensive datasets for training and tend to produce blurry textures when confronted with significant changes in camera viewpoints. Recently, several works have embraced diffusion priors (Liu et al., 2023; Chan et al., 2023; Xu et al., 2023; Gu et al., 2023; Tang et al., 2023; Qian et al., 2023; Chen et al., 2024b) to acquire a probabilistic distribution for unseen views, leading to better synthesis results. Nevertheless, these methods often concentrate on object-centric scenes or lack 3D consistency. Our approach connect 2D images and 3D scenes with explicit 3D Gaussians and incorporate diffusion knowledge, which overcome the mentioned challenges.

## 3 METHOD

Our target is to propose a 3D-aware scene image editing framework (Fig. 2) that allows simultaneous control over the camera and objects. To accomplish this, Sec. 3.1 introduces a novel scene representation called language-guided disentangled Gaussian splatting. In order to achieve object-level control, Sec. 3.2 further distills language features into the Gaussian splatting representation,

achieving disentanglement at the object level. We elaborate the optimization process in Sec. 3.3 and demonstrate the flexible user control enabled by our framework during inference in Sec. 3.4.

### 3.1    3D GAUSSIAN SPLATTING FROM SINGLE IMAGE

**Preliminary.** 3D Gaussian Splatting (3DGS) (Kerbl et al., 2023) has been proved effective in both reconstructive (Luiten et al., 2023; Yang et al., 2023) and generative setting (Zou et al., 2023; Tang et al., 2023). It represents a 3D scene via a set of explicit 3D Gaussians. Each 3D Gaussian describes its location by a center vector $\mathbf{x} \in \mathbb{R}^3$, a scaling factor $\mathbf{s} \in \mathbb{R}^3$, a rotation quaternion $\mathbf{q} \in \mathbb{R}^4$, and also stores an opacity value $\alpha \in \mathbb{R}$ and spherical harmonics (SH) coefficients $\mathbf{c} \in \mathbb{R}^k$ ($k$ represents the degrees of freedom of SH) for volumetric rendering. All the above parameters can be denoted as $\Theta = \{\mathbf{x}_i, \mathbf{s}_i, \mathbf{q}_i, \alpha_i, \mathbf{c}_i | i \in [0, \cdots, N-1]\}$, where $N$ is the number of 3D Gaussians. A tile-based rasterizer is used to render these Gaussians into 2D image.

**Image-to-3DGS initialization.** Given an input image $\mathbf{I} \in \mathbb{R}^{3 \times H \times W}$, an off-the-shelf depth prediction model is applied to estimate its depth map $\mathbf{D} \in \mathbb{R}^{H \times W}$. Then, we could transform image pixels into 3D space, forming the corresponding 3D point clouds:

$$\mathcal{P} = \phi_{2 \to 3}(\mathbf{I}, \mathbf{D}, \mathbf{K}, \mathbf{T}), \tag{1}$$

where $\mathbf{K}$ and $\mathbf{T}$ are camera intrinsic and extrinsic matrices respectively. Such point clouds $\mathcal{P}$ are then used to initialize the 3DGS by directly copying the location and color values, with other GS-related parameters randomly initialized. To refine the 3DGS's appearance, we adopt a reconstruction loss:

$$\mathcal{L}_{\text{recon}} = \|\mathbf{I} - f(\mathcal{P}, \mathbf{K}, \mathbf{T})\|_2^2, \tag{2}$$

where $f$ is the rendering function.

We further enhance the rendered quality by leveraging prior knowledge from image generative foundation model, namely Stable Diffusion (Rombach et al., 2022). It provides update direction to the images rendered by the current 3DGS in the form of Score Distillation Sampling (Poole et al., 2022) loss, denoted as $\mathcal{L}_{\text{SDS}}$.

**3DGS expansion by inpainting.** When camera perspectives changes, rendered views will contain holes due to occlusion or new region outside the original view frustum. We use Stable Diffusion to inpaint the uncovered regions. Then, the newly added pixels need to be accurately transformed into 3D space to align seamlessly with the existing 3D Gaussians.

Previous methods (Chung et al., 2023; Höllein et al., 2023; Yu et al., 2023) first predict the depth values, and then use heuristic methods to adjust the values to align with the existing 3D structure. However, relying on heuristic methods often overlooked various scenarios, leading to artifacts such as depth discontinuities or shape deformations.

Instead, we propose a novel method to lifted novel contents to 3D while ensuring seamless alignment without any heuristic procedures. The key insight is to treat the problem as an image inpainting task, and utilize state-of-the-art diffusion-based depth estimation models (Ke et al., 2024; Fu et al., 2024; Yang et al., 2024) as a prior to solve the task. During denoising steps, rather than using models to predict the noise over the entire image, we employ the forward diffusion process to determine the value of fixed areas (Meng et al., 2021a). This approach guarantees the final result, after denoising, adheres to the depth of original fixed parts, ensuring smooth expansion.

After smooth 3DGS expansion via depth inpainting, we take the imagined novel views as reference views, and apply reconstruction loss $\mathcal{L}_{\text{recon}}$ to supervise the updated 3DGS. SDS loss $\mathcal{L}_{\text{SDS}}$ is adopted for views rendered from camera perspectives that are interpolated between the user-provided viewpoint and the newly imagined views.

### 3.2    LANGUAGE-GUIDED DISENTANGLED GAUSSIAN SPLATTING

Based on the 3DGS built from single input image, users can generate novel views. In this section, we further distill CLIP (Radford et al., 2021) language feature to 3D Gaussians. This introduce semantics into 3D geometry, which helps disentangle individual objects out of the entire scene representation.

**Language feature distillation.** We augment each 3D Gaussian with a language embedding $\mathbf{e} \in \mathbb{R}^C$, where $C$ denotes the number of the channels. Similar to RGB image $\mathbf{I}$, a 2D semantic feature map $\mathbf{E} \in \mathbb{R}^{C \times H \times W}$ can also be rendered by the rasterizer. To learn the embedding, we first use Segment Anything Model (SAM) (Kirillov et al., 2023; Zhang et al., 2023a) to get semantic masks $\mathbf{M}_i$. Then, we can obtain embedding of each object $\mathbf{I} \odot \mathbf{M}_i$ and supervise the corresponding region on rendered feature map $\mathbf{E}$, according to the distillation loss:

$$\mathcal{L}_{\text{distill}} = \sum_i \left\| \left( \mathbf{E} - g\left( \mathbf{I} \odot \mathbf{M}_i \right) \right) \odot \mathbf{M}_i \right\|_2^2, \tag{3}$$

where $g$ is the CLIP's image encoder, and $\odot$ denotes element-wise multiplication. Following LangSplat (Qin et al., 2024), we additionally train an autoencoder to compress the embedding space to optimize the memory consumption of language embedding $\mathbf{e}$.

**Scene decomposition.** After distillation, we can decompose the scene into different objects. This enables user to query and ground specific object, and perform editing over single object (*e.g.* translation, rotation, removal, re-stylizing).

It is worth noting that such scene decomposition property not only enables more flexible edits during inference stage, but also provides augmentation over scene layouts during the optimization process. Since now we can query and render each object independently, we apply random translation, rotation, and removal over objects. This augmentation over the scene layout leads to a significant improvement in the appearance of occluded regions, ultimately enhancing the overall quality of the edited views (see Sec. 4.4).

### 3.3 TRAINING

The overall training objective can be expressed as:

$$\mathcal{L} = \lambda_{\text{recon}} \mathcal{L}_{\text{recon}} + \lambda_{\text{SDS}} \mathcal{L}_{\text{SDS}} + \lambda_{\text{distill}} \mathcal{L}_{\text{distill}}, \tag{4}$$

where $\lambda_{\text{recon}}$, $\lambda_{\text{SDS}}$ and $\lambda_{\text{distill}}$ are coefficients that balance each loss term.

### 3.4 INFERENCE

Due to the disentangled nature of our representation, users can now interact with and manipulate objects in a flexible manner. Here, we mainly discuss prompting objects via two different modalities:

**Text prompt.** Users can query an object through text prompts as shown in Fig. 3. Following LERF (Kerr et al., 2023) and LangSplat (Qin et al., 2024), we calculate the relevancy score `score` between the language embedding $\mathbf{e}$ in the 3D Gaussians and the embedding of the text prompt $\mathbf{e}_l$ as:

$$\texttt{score} = \min_i \frac{\exp(\mathbf{e} \cdot \mathbf{e}_l)}{\exp(\mathbf{e} \cdot \mathbf{e}_l) + \exp(\mathbf{e} \cdot \mathbf{e}_{\text{canon}}^i)}, \tag{5}$$

where $\mathbf{e}_{\text{canon}}^i$ is the CLIP embeddings of canonical phrases including "*object*", "*things*", "*stuff*", and "*texture*". Gaussians that have relevance scores below a predefined threshold are excluded. The remaining part is identified as the object of user interest.

**Bounding box.** Users can also select an object by drawing an approximate bounding box around it on the input image. We group 3D Gaussians within the bounding box by K-Means clustering, and discard clusters whose number of Gaussians does not exceed a threshold proportion.

In the meantime, user can also adjust the camera by specifying intrinsic and extrinsic parameters.

## 4 EXPERIMENTS

### 4.1 SETTINGS

**Implementation details.** To lift an image to 3D, we use GeoWizard (Fu et al., 2024) to estimate its relative depth. Stable Diffusion (Rombach et al., 2022)'s inpainting pipeline is adopted to generate new content for 3DGS's expansion. We leverage MobileSAM (Zhang et al., 2023a) and

Table 1: **User study result.** We report the percentage of favorite users for the consistency and quality of images edited by each method

|  |  | AnyDoor | Ojbect 3DIT | Image Scuplting | Ours |
|---|---|---|---|---|---|
| Consistency | Human | 5.1 | 16.8 | 12.7 | **65.4** |
|  | GPT4-v | 0.0 | 6.7 | 31.3 | **62.0** |
| Quality | Human | 10.4 | 0.5 | 25.1 | **64.0** |
|  | GPT4-v | 6.7 | 13.3 | 39.2 | **40.8** |

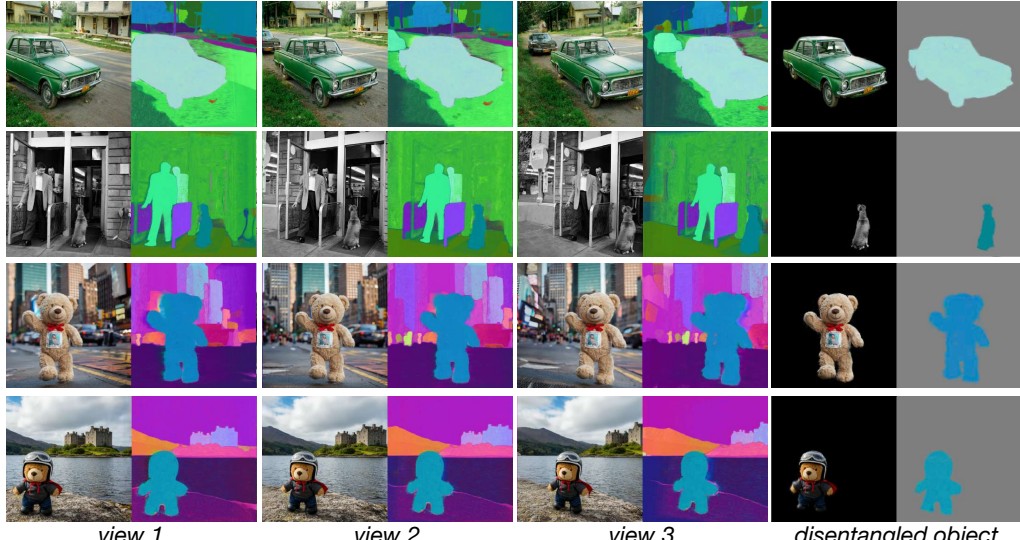

view 1       view 2       view 3       *disentangled object*

Figure 4: **Visualization of rendered images and feature maps.** For each sample, we show three views of rendered images and feature maps. To demonstrate the disentangled scene representation, we use the language embedding to select a foreground object and render it exclusively.

OpenCLIP (Ilharco et al., 2021) to segment and compute rendered views' feature maps, which are further leveraged to supervise the language embedding of 3D Gaussians. We use Stable Diffusion to perform Score Distillation Sampling (Poole et al., 2022) during optimization. Given the already decent image quality at the start of optimization benefited from explicit 3DGS initialization, we adopt a low classifier-free guidance (Ho & Salimans, 2022) scale.

**Baselines.** We compare our method with following scene image editing works: (1) AnyDoor (Chen et al., 2023b) is a 2D diffusion-based model that can teleport target objects into given scene images. It leverages Stable Diffusion's powerful image generative prior by finetuning upon it. (2) Object 3DIT (Michel et al., 2024) is designed for 3D-aware object-centric image editing via language instructions. It finetunes Stable Diffusion over a synthetic dataset containing pairs of original image, language instruction, and edited image. (3) Image Sculpting (Yenphraphai et al., 2024b) is also designed for 3D-aware object-centric image editing. It estimates a 3D model from an object in the input image to enable precise 3D control over the geometry. It also uses Stable Diffusion to refine the edited image quality. (4) AdaMPI (Han et al., 2022) focuses on the control over camera perspective. It leverages monocular depth estimation and color inpainting with learned adaptive layered depth representations. (5) LucidDreamer (Chung et al., 2023) tackles novel view synthesis by querying Stable Diffusion's inpainting pipeline with dense camera trajectories.

## 4.2 QUANTITATIVE RESULTS

We conduct a user study to compare the edited results by our method with the established baselines. We generate 20 samples for each method and request users to vote for their preferred method based on consistency with the original image and quality for each sample. We collect feedback from

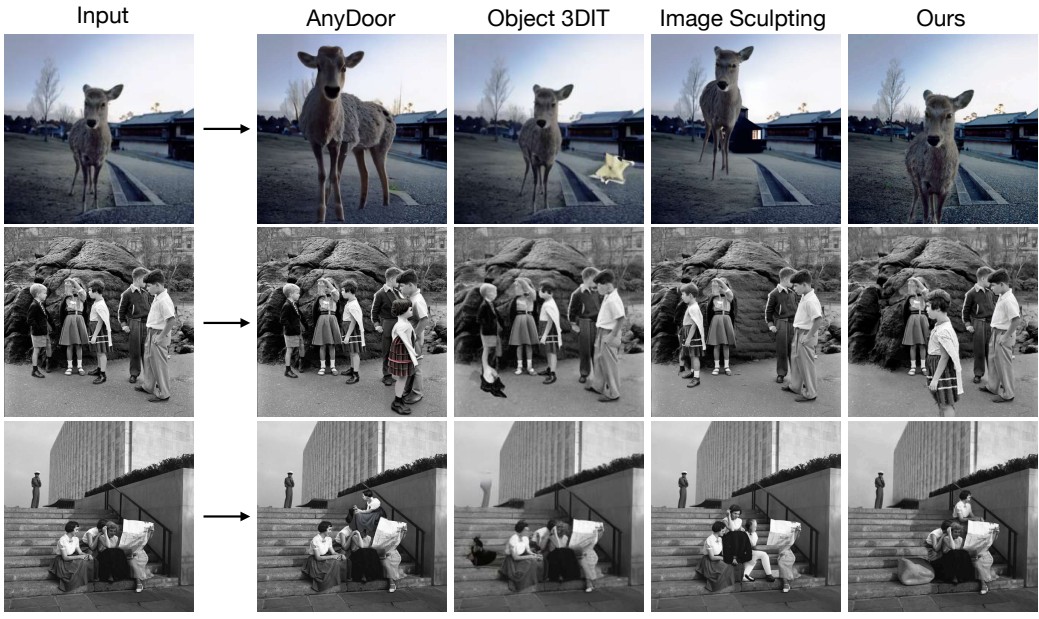

Figure 5: **Comparison results of object-centric manipulation.** We apply translation, resizing, and removal over foreground objects.

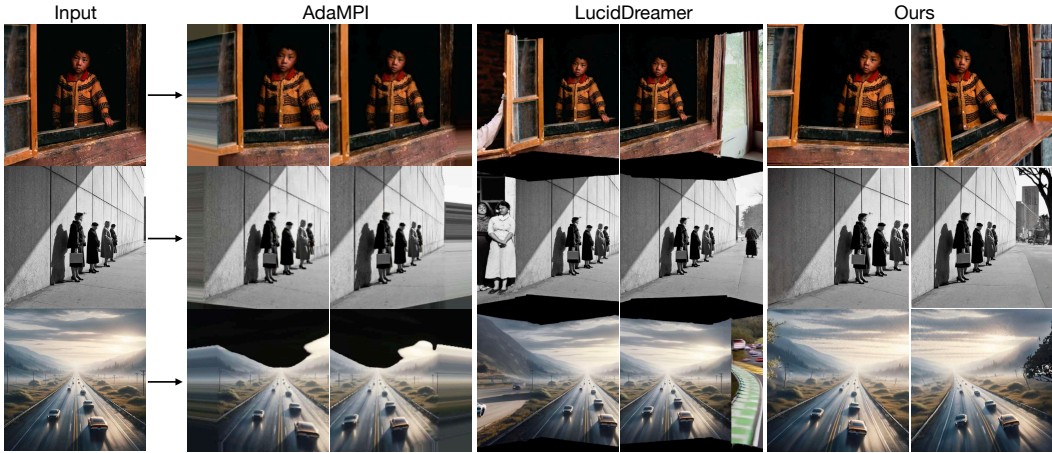

Figure 6: **Comparison results of camera control.** We show two views with different camera perspectives for each method.

25 users, and report the result in Tab. 1. Our method consistently outperforms previous baselines in terms of both consistency and image quality. As recommended in a previous study (Wu et al., 2024b), GPT4-v has the ability to evaluate 3D consistency and image quality. Therefore, we include GPT-4v as an additional criterion. The preference of GPT-4v is well aligned with human preference, which once again demonstrates the superiority of `3DitScene`.

## 4.3 QUALITATIVE RESULTS

Fig. 4 showcases the generated novel views with their respective feature maps produced by our framework. The feature maps demonstrate remarkable accuracy in capturing the semantic content of the images. This ability to distinctly separate semantic information plays a crucial role in achieving precise object-level control In the following, we demonstrate flexible editing over scene images enabled by our framework, and also compare with baseline methods.

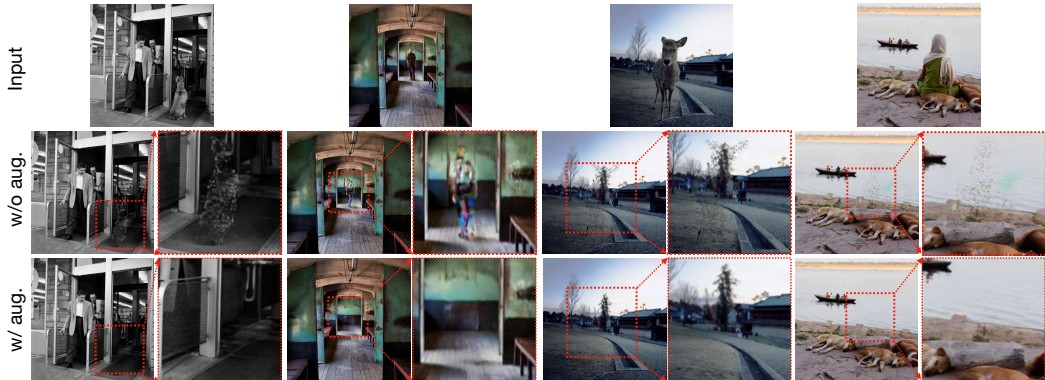

Figure 7: **Ablation results for layout augmentation during optimization.** To evaluate the degree of object-level disentanglement, we conduct object removal for each sample. The top row displays the input image, while the next two rows showcase the edited scene

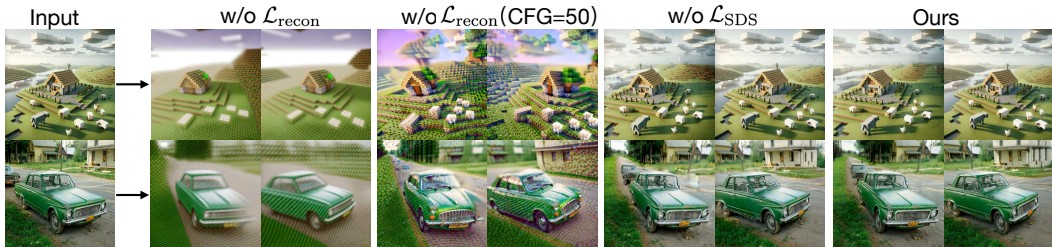

Figure 8: **Ablation results for loss terms.** We show rendered novel views under different loss settings. The left column lists the input image. In right columns, two views are shown for each configuration. The quality degrades when reconstruction or SDS loss term is discarded

**Object manipulation.** Since different methods define object manipulation, particularly translation operations, in different coordinate systems[1], it becomes challenging to evaluate them under a unified and fair setting. Therefore, we evaluate each method under its own specific setting to achieve the best possible result. As shown in Fig. 5, AnyDoor struggles to maintain object identity and 3D consistency when manipulating object layouts, primarily due to the absence of 3D cues. Object 3DIT, trained on synthetic datasets, exhibits limited generalization ability to real images. By leveraging a 3D model derived from the input image, Image Sculpting achieves better results. Nonetheless, it encounters issues with inconsistency when manipulating objects.

In contrast, our method delivers satisfactory 3D-aware object-level editing results. It maintains accurate 3D consistency of edited objects after rearranging their layout. Additionally, it preserves occlusion relationships within the scene, such as moving the girl to be partially occluded by a foreground object in the last row example.

**Camera control.** We compare our methods with AdaMPI and LucidDreamer for camera control. As illustrated in Fig. 6, AdaMPI only focuses on scenarios where the camera zooms in, and does not consider novel view synthesis. Therefore, this approach is not suitable for 3D-aware image editing when large camera control is required. LucidDreamer also leverages Stable Diffusion's inpainting capacity for novel view synthesis. However, it suffers from sudden transitions in the content within the frame (see sample in the bottom line). It also requires dense camera poses. In contrast, our method only needs as few as three camera poses and enables smooth transitions from the input view to novel views, enhancing user control over the camera perspective.

---

[1]AnyDoor, Object 3DIT and Image Sculpting respectively employs 2D masks, language prompts, and image coordinates for control. We use coordinates in 3D space instead.

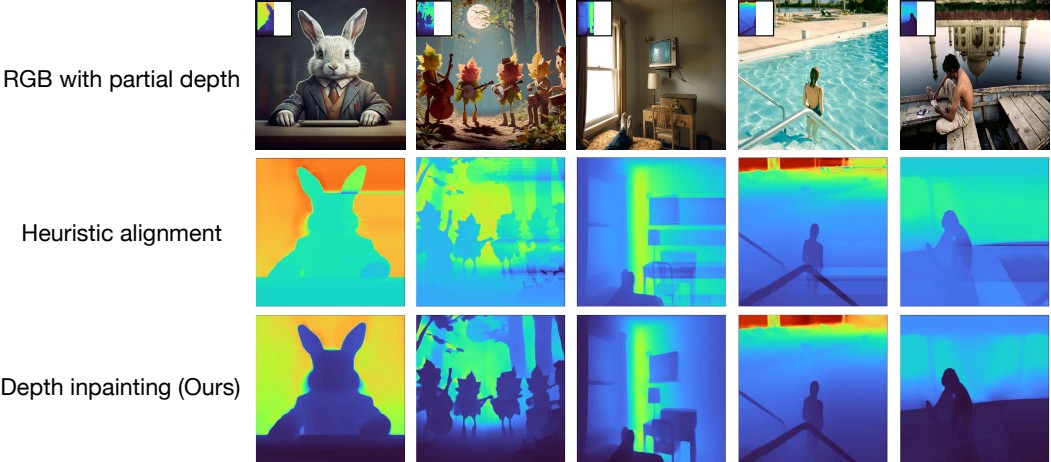

Figure 9:  **Ablation results for depth inpainting.**  The first row shows images with their corresponding depth maps (available on the left half). The second and third row display the depth map predicted by heuristic alignment, and our depth inpainting method respectively.

## 4.4 ABLATION STUDY

**Layout augmentation during optimization.**    As our representation disentangles at object level, we could perform layout augmentation during optimization. Here, we investigate whether disentanglement property benefits the optimization process. We use the task of removing objects to evaluate the degree of disentanglement.

As illustrated in Fig. 7, when layout augmentation is disabled during optimization, floating artifacts can be observed. We discover that these Gaussians lie inside the object. They are occluded by Gaussians at the surface. As they do not contribute to the rendering result, they are consequently not updated by gradient descent during optimization, leaving their language embeddings unsupervised.

In contrast, when applying layout augmentation during optimization, such Gaussians will be exposed when the foreground object is moved away, and hence updated. With this ablation, it is concluded that the disentanglement property of the proposed representation not only enables more flexible inference, but also contributes to the optimization process.

**Loss terms.** During optimization, we adopt three loss terms: $\mathcal{L}_{\mathrm{recon}}$, $\mathcal{L}_{\mathrm{SDS}}$, and $\mathcal{L}_{\mathrm{distill}}$. $\mathcal{L}_{\mathrm{distill}}$ plays a critical role in distilling language embedding into 3D. The remaining two terms focus on enhancing the visual quality of images. As illustrated in Fig. 8, the image quality degrades severely without $\mathcal{L}_{\mathrm{recon}}$ or $\mathcal{L}_{\mathrm{SDS}}$. Without $\mathcal{L}_{\mathrm{recon}}$, the image is only refined by the SDS loss, which creates discrepancies with the original image. When the CFG value is set low, 5 as default, the image appears lacking in details and exhibits unusual texture patterns. Increasing the CFG value introduces more details, yet leads to inconsistencies with the original image, while the issue of strange texture patterns persists. Additionally, only applying $\mathcal{L}_{\mathrm{recon}}$ results to floating artifacts and blurriness across the entire image. In conclusion, both SDS and reconstruction loss are crucial for achieving decent image quality.

**Depth inpainting.** When expanding 3DGS at novel views, we need to estimate the depth map of unseen regions. Here, we compare our inpainting-based depth estimation with heuristic-based method. Fig. 9 show images with depth map available in the left part. The task is to predict the depth map of the right part. Method relying on heuristic alignment results to artifacts like depth discontinuity. In contrast, our proposed method is capable of producing accurate depth maps that align well with the left known part.

## 5 CONCLUSION AND DISCUSSION

We present a novel framework, 3DitScene, for scene image editing. Our primary objective is to facilitate 3D-aware editing of both objects and the entire scene within a unified framework.

We achieve this by leveraging a new scene representation, language-guided disentangled scene representation. This representation is learnt by distilling CLIP's language feature into 3D Gaussians. The semantic 3D Gaussians effectively disentangle individual objects out of the entire scene, , thereby enabling localized object editing. We test `3DitScene` under different settings and prove its superiority compared to previous methods.

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

APPENDIX

## A  IMPLEMENTATION DETAILS

**Scene initialization.**  We first utilize GeoWizard (Fu et al., 2024) to estimate the relative depth of the input image. Next, we lift the image into 3D space based on this depth and perform 300 SDS steps. The camera azimuth angle is randomly sampled from $[-15°, 15°]$. After 300 steps, we employ Stable Diffusion (Rombach et al., 2022)'s inpainting pipeline to generate new content for the expansion of 3DGS. Specifically, we inpaint the rendered views at azimuth angles of $-15°$ and $15°$. Finally, we use GeoWizard (Fu et al., 2024) again to estimate the depth of the newly added regions and lift them into 3D space.

**SDS optimization.**  We perform 1500 SDS steps to optimize the whole scene. We randomly sample the diffusion time step from $[l, r]$, where $l = 0.02$, and $r$ starts at $0.5$ and gradually decreases to $0.2$ by the 1000th step. We use guidance strength of 5 for classifier-free guidance.

**Coefficients.**  In Eq. (4), we choose $\lambda_{\text{recon}} = 1000$, $\lambda_{\text{SDS}} = 0.01$, and $\lambda_{\text{distill}} = 1$.

## B  EDITING PIPELINE AND USER INTERFACE

To enhance the user experience, we explore two types of editing frameworks:

1. **Large language model (LLM) based.**  We prompt the LLM to take the user's editing request as input and parse it into predefined components, including camera movement angles, descriptions of the object of interest, and the transformation matrix for that object. Our editing algorithm then uses these components as input to optimize the 3D scene and carry out the necessary edits.

   However, relying solely on the LLM as the interface for our editing algorithm has its drawbacks; it makes it difficult for users to interactively manipulate and edit the optimized 3D scene. To address this, we have also developed a user interface that facilitates these interactions.

2. **User interface (UI) based.**  We developed an user interface for editing.  We have implemented a web-based interactive panel that allows users to control the editing process. Fig. A1 provides an overview of the user interface. To use it, users need to upload an image, specify the object of interest, and provide text prompts, as illustrated in Fig. A2. They can then visualize the optimization process in real time.  After the optimization is complete, users can use draggable sliders for various edits, including moving, removing, and rotating objects, as shown in Fig. A3, Fig. A4, and Fig. A5.

Both types of editing workflows do not require users to interact with the code during the entire process.

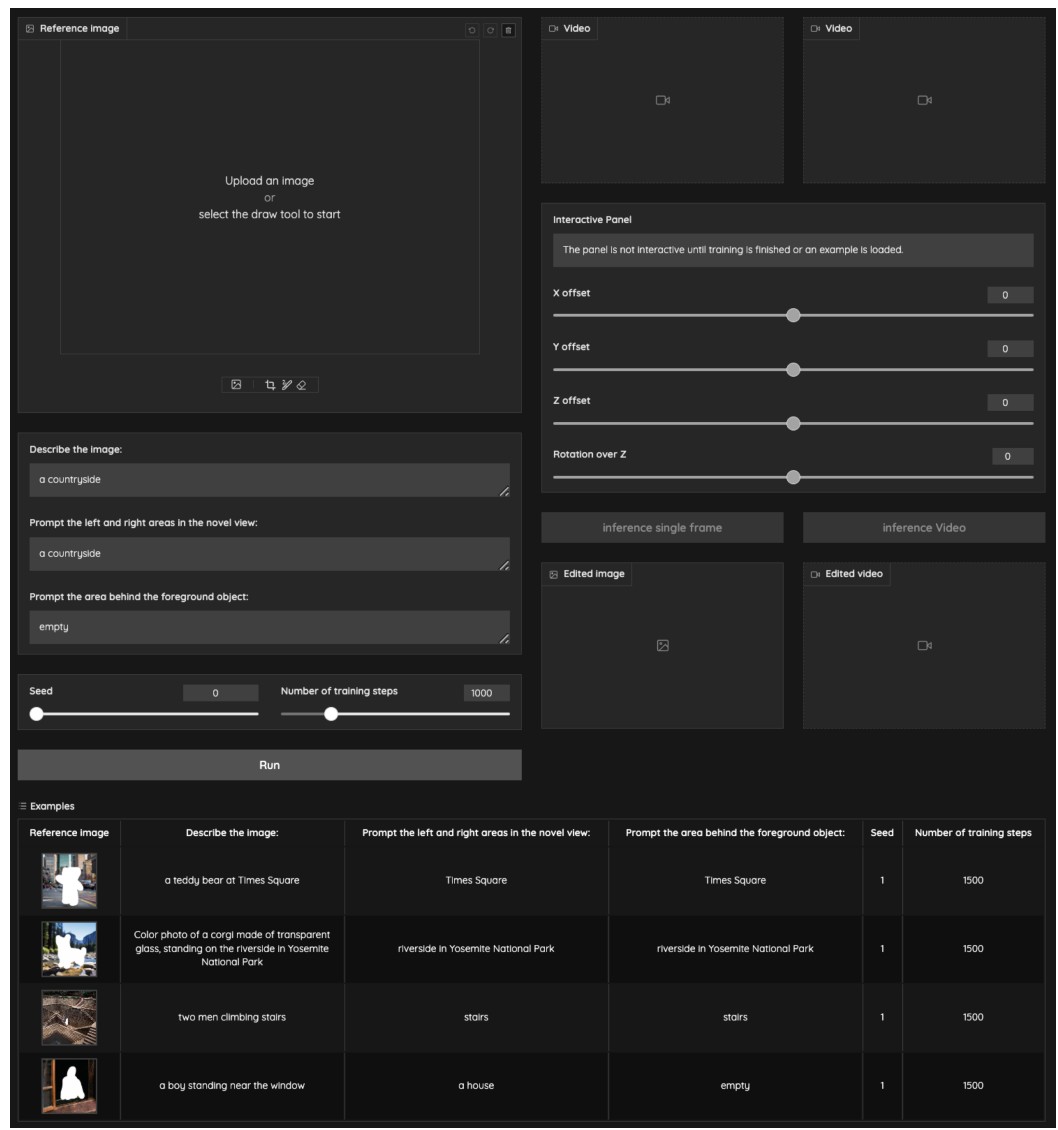

Figure A1: **Overview of the user interface.** To provide better user experience, we have developed a web-based interface. Users simply need to upload the input image and enter a text prompt. They can then visualize the optimization process in real time and edit the scene using sliders.

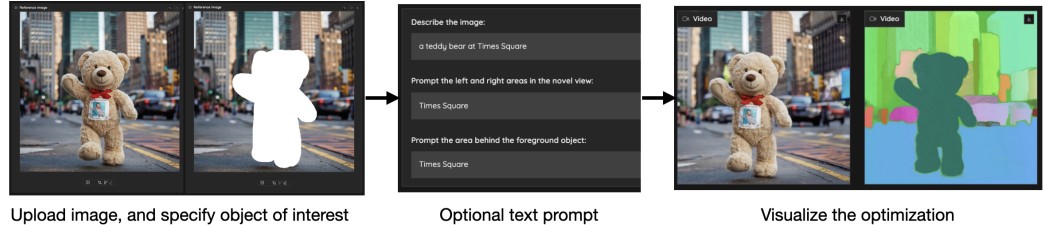

Figure A2: **Procedure to optimize a scene via the user interface.** To optimize a 3D scene for 3D-aware image editing, users need to upload an image, specify the object of interest, and provide text prompts. They can then visualize the optimization process in real time.

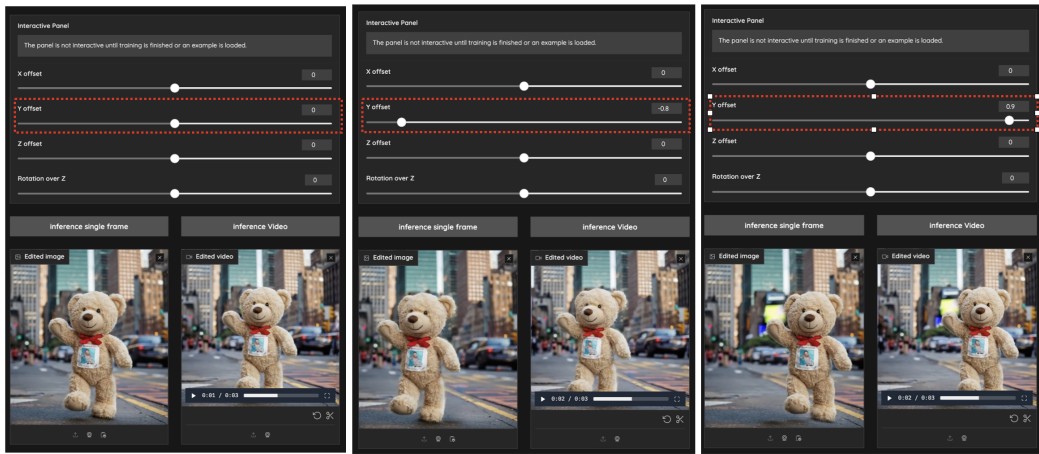

Figure A3: **Interactive editing via the user interface.** In this sample, we demonstrate that by sliding the bar, users can adjust the object's offset along the Y coordinate.

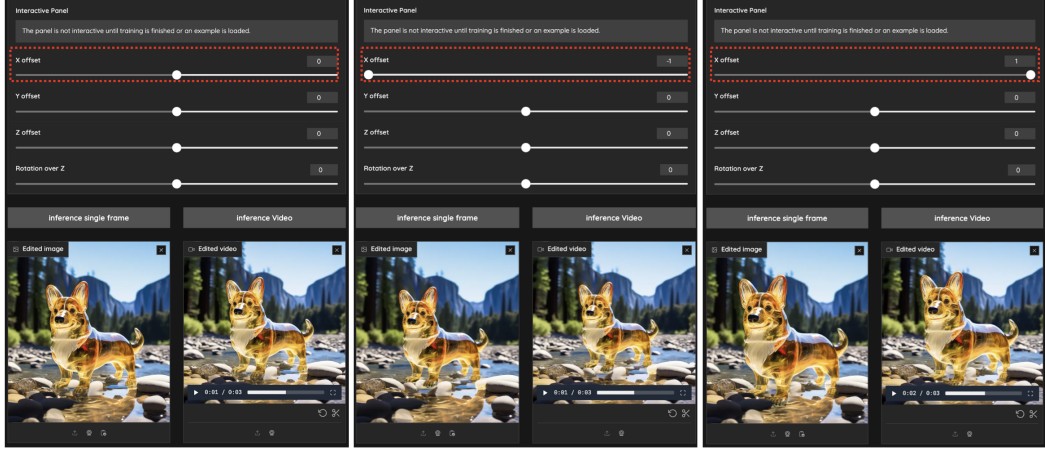

Figure A4: **Interactive editing via the user interface.** In this sample, we demonstrate that by sliding the bar, users can adjust the object's offset along the X coordinate.

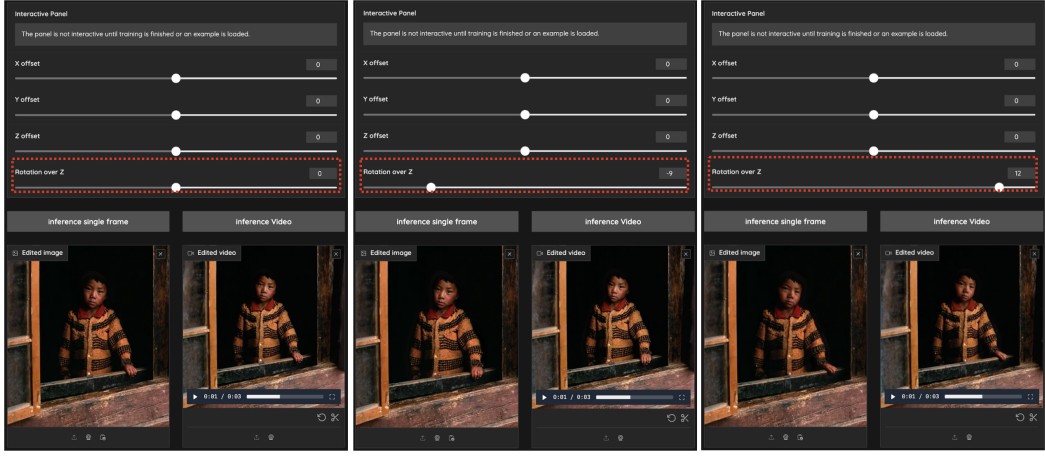

Figure A5: **Interactive editing via the user interface.** In this sample, we demonstrate that by sliding the bar, users can adjust the object's rotation.

