# OpenReview forum: "3DitScene: Editing Any Scene via Language-guided Disentangled Gaussian Splatting"
_ICLR.cc/2025/Conference — ICLR 2025 Poster_

### Official Review · Reviewer_fiMW · 2024-11-03

**Soundness:** 3
**Presentation:** 3
**Contribution:** 2
**Rating:** 5
**Confidence:** 3

**Summary:**

The paper proposed a unified framework, 3DitScene, for 3D-aware 2D image editing with different level of granulairies and diverse editing tasks, such as object level editing and camera adjustments.
By converting a single image into 3D Gaussian splatting (3DGS) representation, 3DitScene generates additional 3D content in previously unseen areas via RGB and depth inpainting from diffussion priors. Additionally, it distills CLIP language semantics into the expanded 3D. With the recontructed 3D scene with disentangled 3D semantics, the framework enables editing of both viewing camera and objects in the the image by camera, bounding box and textual controls.

**Strengths:**

1. The work proposed a 3D-aware image editing pipeline by establishing semantic 3D representation via 2D-to-3D inpainting for novel view completion and language semantics lifting for semantic disenanglements. Under this language-guided disenangled 3D representation, the editing supports object-level maipulatiion and camera controls.

2. Evaluation experiments are extensive for performance and comparisons with baselines. The method designs are validated through conducting ablation studies, exploring various alternative designs, including layout augmentation, losses, depth inpainting strategies.

**Weaknesses:**

1. While the paper proposed a practical and accurate image editing system, it does not demonstrate the editing procedure for practical use. Also see Questions 1 for further details.

2. Some details are missing for method reproduction and training, such as trainining parameter configuration. It is also unclear about camera configuration, e.g. whether K and T in Eq.1 are always preset and fixed for given single image? More details are needed to facilitate reproduction and editing controls.

Additional concerns can be found in the Questions.

**Questions:**

1. The paper lacks clarity on the editing process, despite mentioning text, bounding boxes, and camera controls in Sec.3.4.
The detailed operation steps are still ambiguous, and there is no clear editing procedure outlined in paper or video demo.

    From Sec.3.4, it appears that editing primarily involves "text-guided" editing by querying specific object + object manipulation?

    Is "camera control" achieved always by changing camera instrincs and extrinsics parameters? But in Fig.1, it seems that text prompts are also used for control.

    In addition, there's no examples of bounding box usage in the paper.

    I also suggest more editing details such as text prompts in Fig.5 and 6 and video, similar to what is shown in Fig.1.

2. Since the 3D-aware editing is based on explicit 3DGS, and the disenangled object manipulation does not involve whole image re-generation. Accordingly, lighting inconsistencies arise after editng, e.g., shadows do not change appropriately as object moves. Can the lifted semantics help resolve this problem, or do you have any ideas can handle this obvious artifact?

3. Language-semantics 2D-to-3D lifting is increasingly common nowadays, e.g., LERF, LangSplat, Gaussian Grouping. What are the advantages of proposed semantics lifting schemes over these existing works? Additionally, would other language-embedded 2D-to-3D lifting methods should be compatible with this proposed editing framework?

---

> ### Author Response · Authors · 2024-11-25
>
> Q1: The paper does not demonstrate the editing procedure for practical use.
>
> A1: Thanks for pointing this out. We have included an additional section to introduce the editing procedure in the Appendix. To ease and automate the procedure, we have developed an user interface for user to interact with our algorithm.
>
> Q2: Missing details.
>
> A2: We have included additional implementation details in the supplementary materials. Furthermore, we will release the code as soon as this work gets accepted, as we believe this will greatly assist the audience in using and reproducing our work. For camera configuration, $K, T$ are fixed during scene initialization stage.
>
> Q3: it appears that editing primarily involves "text-guided" editing by querying specific object + object manipulation?
>
> A3: Yes. We made an illustration (Figure A1-2) in the Appendix to show how the editing works.
>
>
> Q4: Is "camera control" achieved always by changing camera instrincs and extrinsics parameters? But in Fig.1, it seems that text prompts are also used for control.
>
> A4: Camera control is primarily achieved by adjusting the camera intrinsics and extrinsics. However, these factors do not need to be directly exposed to the user. Instead, we utilize the LLM to interpret users' text prompts and convert them into the necessary camera parameters.
>
>
> Q5: In addition, there's no examples of bounding box usage in the paper.
>
> A5: We provide examples of bounding boxes (user-created rough painting masks) in the Appendix.
>
>
> Q6: Since the 3D-aware editing is based on explicit 3DGS, and the disenangled object manipulation does not involve whole image re-generation.
>
> A6: 3D Gaussian Splatting does not explicitly model shadows and lighting; instead, it bakes them into the appearance. Consequently, our method, which uses 3DGS as the underlying representation, also lacks the ability to model these lighting factors. One potential solution is to consider regenerating the entire image, for instance, by applying SDS steps across the whole region. Another promising direction is to explore 3DGS with explicit lighting modeling.
>
> Q7: What are the advantages of proposed semantics lifting schemes over existing language-semantics 2D-to-3D lifting works?
>
> A7: Existing works such as LERF, LangSplat, and Gaussian Grouping begin with a well-optimized 3DGS and then lift 2D semantics to 3D. In contrast, our work, which focuses on image editing, starts from a single image. We show that 2D-to-3D semantics lifting can effectively lead to high-quality 3D-aware image editing. We hope that this could

---

> ### Author Response · Authors · 2024-11-30
>
> As the rebuttal deadline approaches, we would like to know if you have any further concerns regarding our work. We would appreciate the opportunity to clarify any issues. If there are no additional concerns, could you please kindly consider updating the score accordingly? Thank you!

---

> > ### Comment · Reviewer_fiMW · 2024-12-03
> >
> > Thank you for your response. After reviewing the authors' replies, the revision, and the other reviews, I appreciate the editing results and the proposed general editing tasks. I believe this work represents a solid engineering effort.
> >
> > However, I still have concerns regarding the limited technical contribution, and I feel that the main paper, particularly the implementation details, should be further enhanced. Therefore, I would like to maintain my original score.

---

### Official Review · Reviewer_Fas1 · 2024-11-03

**Soundness:** 3
**Presentation:** 3
**Contribution:** 2
**Rating:** 5
**Confidence:** 2

**Summary:**

The authors introduce 3DitScene, a unified framework that employs language-guided disentangled Gaussian Splatting, allowing seamless transitions from 2D to 3D editing. This approach utilizes refined 3D Gaussians informed by CLIP’s language features, enabling precise control over both individual objects and overall scene composition. Experimental results validate the effectiveness and versatility of 3DitScene, demonstrating its superiority over existing methods.

**Strengths:**

1. 3DitScene effectively integrates various editing requirements into a single framework, facilitating comprehensive manipulation of both scenes and objects.
2. The framework employs language-guided disentangled Gaussian Splatting to produce a detailed 3D scene representation, which enables novel view synthesis and improves object decomposition.

**Weaknesses:**

The implementation details are not clearly articulated. Specifically, the training and optimization specifics for 3D Gaussian Splatting (3DGS) are lacking, including the number of iterations during initialization and for novel view optimization. Furthermore, the parameters of the loss function, such as λrecon, are not specified. Additionally, the experimental results do not appear to be satisfactory.

**Questions:**

See what is written in the weaknesses section.

---

> ### Author Response · Authors · 2024-11-25
>
> Q1: The implementation details are not clearly articulated.
>
> A1: We have included additional implementation details in the supplementary materials. Furthermore, we will release the code as soon as this work gets accepted, as we believe this will greatly assist the audience in using and reproducing our work.

---

> ### Comment · Reviewer_Fas1 · 2024-11-26
>
> Thank you for your response~

---

> > ### Author Response · Authors · 2024-11-27
> >
> > Thank you for taking the time to review our response. We have included implementation details in the Appendix, such as the number of iterations for different stage, and the loss function coefficients. Do you have any further concerns regarding our work? If not, would you consider increasing your score?

---

### Official Review · Reviewer_gw3n · 2024-11-05

**Soundness:** 3
**Presentation:** 3
**Contribution:** 2
**Rating:** 5
**Confidence:** 4

**Summary:**

This paper presents a scene editing framework that leverages language-guided disentangled Gaussian Splatting. Initially, the input image is projected into 3D Gaussians, which are subsequently refined and enriched through a 2D generative prior. Language features from CLIP are then embedded into the corresponding 3D Gaussians. This representation enables view changes and text-guided editing of the scene.

**Strengths:**

The proposed technique is well-founded, and the experimental results are impressive.

**Weaknesses:**

The novelty appears limited, as the proposed method largely combines existing techniques. Projecting a 2D image into 3D Gaussians based on depth information has been demonstrated in works like LucidDreamer, among others. Additionally, embedding language features into 3D Gaussian Splatting (3DGS) and enabling text-guided editing have been explored in methods such as Grouping Gaussian, LangSplat, and GaussianEditor.

It is unclear how the proposed method outperforms prior techniques, such as LucidDreamer and AdaMPI, in Figure 6. While LucidDreamer and AdaMPI exhibit some out-of-range areas, simple inpainting could potentially address these issues.

Since the method relies on depth maps to generate 3D Gaussian splats, it appears limited in its ability to handle significant changes in view angle.

The method seems to only function with generated images rather than real-world inputs.

**Questions:**

Please refer to the weaknesses section.

---

> ### Author Response · Authors · 2024-11-25
>
> Q1: Limited novelty. The proposed method largely combines existing techniques.
>
> A1: We agree with the point that the techniques used in our work originates from previous works. However, we would like to emphasize that these techniques were originally designed for different tasks—such as LucidDreamer for image-to-3D conversion without editing capabilities, and DreamFusion for text-to-3D generation—which differ significantly from the application and task that we focus. We believe the value of our work lies in demonstrating a viable approach to unify these techniques for 3D-aware image editing. Additionally, our work can continue to benefit from advancements in these areas in the future.
>
> Q2: It is unclear how the proposed method outperforms prior techniques, such as LucidDreamer and AdaMPI.
>
> A2: While prior works like LucidDreamer and AdaMPI can potentially generate novel view content through inpainting, they do not support 3D-aware object-level editing. In contrast, our work enables both camera control and object editing within a unified framework.
>
> Q3: Since the method relies on depth maps to generate 3D Gaussian splats, it appears limited in its ability to handle significant changes in view angle.
>
> A3: We use depth estimation only as a rough initialization, with the scene geometry refined during SDS optimization. Additionally, our method can be easily adapted to accommodate significant changes in view angle. We can progressively expand the azimuth range and inpaint novel views according to a schedule (for example, [−15,15] for the first 300 steps, followed by [−30,30] for the next 300 steps, and so on). In terms of image editing, we believe current method can meet the most demands of users and be further improved in future.
>
> Q4: The method seems to only function with generated images rather than real-world inputs.
>
> A4: Our work works well with real-world images (see the first two samples of Figure 4, all three samples of Figure 5, first two samples of Figure 6, and all the samples of Figure 7).

---

> ### Author Response · Authors · 2024-11-30
>
> As the rebuttal deadline approaches, we would like to know if you have any further concerns regarding our work. We would appreciate the opportunity to clarify any issues. If there are no additional concerns, could you please kindly consider updating the score accordingly? Thank you!

---

### Official Review · Reviewer_d3t4 · 2024-11-05

**Soundness:** 3
**Presentation:** 3
**Contribution:** 4
**Rating:** 8
**Confidence:** 3

**Summary:**

This paper provides a method for highly precise 3d scene editing.
The author starts from a single image input, first using estimated depth to init the 3dgs. Use inpainting for both novel view and its depth map. Authors also introduce a language feature distillation loss, so that we will be able to use text to query the gs during inference

**Strengths:**

The quality is very well. The proposed methods are solid

Authors also conduct a lot of  experiments to compare with multiple baselines.

**Weaknesses:**

I am afraid that the SAM mask is not accurate, the distill loss may cause some artifacts. Like some of the gaussians in the edge may not be able to match the image prompt. Is it true?
Current method cannot deal with shadow and lightning, like the Rotate the camera then delete the dog case in the teaser
Need more visual results.

**Questions:**

How long does it take for training one scene?
How long does it take for rendering one image?
I don't understand why “Remove the headscarf then move the camera” case in the teaser can generate the hair of the woman. cause in inference you just manipulate the gs, is ther anything generative that enables this capability?

I pretty much love the paper, if the author can provide more details, I am happy to raise my score.

---

> ### Author Response · Authors · 2024-11-25
>
> Q1: I am afraid that the SAM mask is not accurate, the distill loss may cause some artifacts. Like some of the gaussians in the edge may not be able to match the image prompt.
>
> A1: While it’s true that the SAM mask can be inaccurate, this does not lead to artifacts. With our proposed Layout Augmentation, any mismatched Gaussians at the edges will not move with the object, allowing them to be pruned during optimization. Figure 7 clearly demonstrates the ablation study for Layout Augmentation. Without this, the inaccurate mask can indeed cause floating artifacts, but Layout Augmentation effectively resolves this issue.
>
> Q2: Current method cannot deal with shadow and lighting.
>
> A2: Yes, this is true. 3D Gaussian Splatting does not explicitly model shadows and lighting; instead, it bakes them into the appearance. Consequently, our method, which uses 3DGS as the underlying representation, also lacks the ability to model these lighting factors. Although explicitly modeling lighting in 3DGS is a promising direction for future research, it is beyond the scope of our current work. We believe our approach could greatly benefit from advancements in this area.
>
>
> Q3: How long does it take for training one scene? How long does it take for rendering one image?
>
> A3: Training a single scene takes 10 minutes. We can achieve a rendering speed of 30 frames per second.
>
> Q4: Is ther anything generative that enables this capability?
>
> A4: As mentioned in lines 183-186, we utilize Stable Diffusion to inpaint the unseen regions, which may arise from novel views or occlusions. Specifically, we inpaint the occluded areas behind the object of interest with the generative prior. This is why the case of “removing the headscarf and then moving the camera” in the teaser can generate the woman's hair.

---

> > ### Comment · Reviewer_d3t4 · 2024-11-29
> >
> > Thank you so much for your response! my concerns are resolved. I have raised my score to 8

---

> > > ### Author Response · Authors · 2024-11-30
> > >
> > > We thank the reviewer for recognizing our explanation.

---

### Official Review · Reviewer_txJu · 2024-11-08

**Soundness:** 2
**Presentation:** 3
**Contribution:** 3
**Rating:** 6
**Confidence:** 4

**Summary:**

The authors present a new image editing pipeline that lift 2D data priors, including CLIP, mono-depth, SAM and Stable Diffusion, into a 3D Gaussian Splatting representation for manipulation. This decomposes the scene into a camera and queryable objects that can be rotated and translated in 3D space. The process introduces discontinuities at novel views, which the authors propose to fill in following an image inpainting procedure and using the SDS loss from Dreamfusion. The resulting scene can be rendered to produce an edited image.

**Strengths:**

Strengths:
* The authors leverage a broad variety of methods to extend image features to 3D, all of which are well-justified inclusions.
* Authors ablate the main technical decisions in the paper, specifically the different loss terms, CFG scale and object removal.
* Uses GPT4o LLM-as-critic evaluations from past works to automatically evaluate 3D generation quality.
* Writing is focused and practical, highlighting different elements of the design space and clearly explaining each decision the authors made.
* Method is training-free, so it should improve over time as base models improve.
* Figures are well-presented and understandable.
* The importance of the proposed task is clear and the method seems to be a reasonable improvement over existing works.

**Weaknesses:**

Weaknesses:
* The main weakness of this paper is that it mostly combines existing techniques in a pipeline to improve an application rather than introducing new technical insights. For instance:
  * LucidDreamer used monocular depth to unproject an image into a Gaussian Splatting scene
  * LERF introduced language-embedded 3D representations
  * Dreamfusion introduced the SDS objective
  * Inpainting techniques are already broadly explored in the image diffusion model literature
* Evaluations are very limited, with the user study not being well-explained and including a small number of participants (25) and generated samples (20). This is a known challenge in this problem space, but it would have been nice to see more discussion or contribution in this direction.

**Questions:**

* What is the runtime of the proposed method and other alternatives? While the 3D representation seems to improve quality, it seems unlikely that this multi-stage method would be fast enough for real use.
* The teaser figure seems to suggest that this process is "hands-off," taking in a text prompt about manipulating the scene and then generating, but I'm having trouble telling if this is actually true. Could the authors please clarify this?
* To what extent does the method begin to fail if there are large baseline camera movements? SDS at low CFG has very limited generation ability, for instance.
* Are there other downstream tasks where this pipeline could be useful? Maybe other instances of image editing?

---

> ### Author Response · Authors · 2024-11-25
>
> Q1: The main weakness of this paper is that it mostly combines existing techniques in a pipeline to improve an application rather than introducing new technical insights.
>
> A1: We agree with the point that the techniques used in our work originates from previous works. However, we would like to emphasize that these techniques were originally designed for different tasks—such as LucidDreamer for image-to-3D conversion without editing capabilities, and DreamFusion for text-to-3D generation—which differ significantly from the application and task that we focus. We believe the value of our work lies in demonstrating a viable approach to unify these techniques for 3D-aware image editing.
>
> Q2: Evaluations are very limited.
>
> A2: We acknowledge that the number of participants in our human study is small; however, we want to emphasize that the results are significant, which clearly demonstrate our performance improvement over previous methods. We also agree with your point about the absence of an objective metric for measuring image editing, as it is a complex and subjective task. We believe this is an interesting direction worth exploring in the future.
>
> Q3: What is the runtime of the proposed method and other alternatives?
>
> A3: Our method requires 10 minutes to process a scene, while most alternatives, such as AnyDoor and Object 3DIT, are feed-forward methods that take significantly less time (less than one minute). However, these alternatives do not achieve precise 3D-related editing. To reduce the time cost of our method, it would be beneficial to explore the use of amortization to incorporate per-scene optimization into a generalizable feed-forward model.
>
> Q4: The teaser figure seems to suggest that this process is "hands-off”.
>
> A4: Yes, the process is quite user-friendly. We illustrate the whole process in the Appendix. We explore two methods to achieve this: (1) using a large language model (LLM). We prompt the LLM to take the user's editing request as input and parse it into predefined components, such as camera movement angles, descriptions of the object of interest, and the transformation matrix for that object; (2) providing an interactive panel for refining the edits. We have implemented a web-based interactive panel that allows users to control the editing process. Draggable sliders facilitate various edits, including moving, removing, and rotating objects, so users do not need to interact with the code throughout the entire editing workflow.
>
> Q5: To what extent does the method begin to fail if there are large baseline camera movements?
>
> A5: Our method can be easily adapted to accommodate large baseline camera movements. We can progressively expand the azimuth range and inpaint novel views according to a schedule (for example, [−15,15] for the first 300 steps, followed by [−30,30] for the next 300 steps, and so on). However, since our primary focus is on image editing, we believe that significantly altering camera poses is outside the scope of our work, and therefore we do not emphasize this capability in the paper.
>
> Q6: Are there other downstream tasks where this pipeline could be useful? Maybe other instances of image editing?
>
> A6: Our method enables basic 3D editing capabilities, including translation, rotation, removal, and resizing, for image editing. Additionally, it can be integrated with other image-based or 3DGS-based editing techniques. For instance, we can combine our approach with GaussCtrl [1], ProEdit [2], or MvDrag3D [3]  to alter the shape of an object within a given image.
>
> [1] GaussCtrl: Multi-View Consistent Text-Driven 3D Gaussian Splatting Editing
> [2] ProEdit: Simple Progression is All You Need for High-Quality 3D Scene Editing
> [3] MvDrag3D: Drag-based Creative 3D Editing via Multi-view Generation-Reconstruction Priors

---

> > ### Comment · Reviewer_txJu · 2024-11-28
> >
> > Thank you for your response.

---

> > > ### Author Response · Authors · 2024-11-30
> > >
> > > We thank the reviewer for the timely reply and helping improve the draft. Please let us know if there are any additional questions we can further address.

---

### Meta-Review · Area_Chair_hKgi · 2024-12-21

**Metareview:**

The paper introduces a new framework for 3D scene editing that leverages language-guided Gaussian splatting. Reviewers generally praised the quality of the editing results and the potential of the approach, but also raised concerns about the limited novelty, as the method primarily combines existing techniques.

Although scores remain mixed, the main complaints seem to be about technical novelty---and in my eyes, combining a set of existing methods into a novel system/application seems like a sufficiently significant contribution for publication.

**Additional Comments On Reviewer Discussion:**

Scores were initially mixed, with mostly borderline rejects and one solid accept.  The reviewers addressed a handful of reviewer concerns in their responses, but the majority reviewers did not seem to be convinced by the discussion and kept their original scores.

The main complaints about the paper seem to be about novelty, but from a survey of the proposed method, despite being a combination of previously explored ideas, seems sufficiently novel in its combination of multiple pieces.

---

### Decision · Program_Chairs · 2025-01-22

Accept (Poster)